# The Spreading in Europe of the Non-Indigenous Species *Oenothera speciosa* Nutt. Might Be a Threat to the Autochthonous Moth *Macroglossum stellatarum* (Linnaeus, 1758)? A New Case Study from Italy

**Andrea Bonifazi** [1,*] 🄳, **Marta Pacini** [1] **and Emanuele Mancini** [2]

[1] ARPA Lazio, Agenzia Regionale per la Protezione Ambientale del Lazio, Dipartimento Stato dell'Ambiente, 00173 Rome, Italy
[2] Italian Fishery Research and Studies Center, 00184 Rome, Italy
* Correspondence: andrea.bonifazi@arpalazio.it

**Abstract:** *Oenothera speciosa* Nutt. is a non-indigenous plant that is widespread in Europe, South America, Asia, and Oceania. Although in its native range it is rarely pollinated by sphingid moths, in Europe and Asia, it was found to be associated with the hummingbird hawkmoth *Macroglossum stellatarum* (Linnaeus, 1758). However, the plant–insect interaction was negative, and the moths were found with proboscides stuck to the flowers of this plant. This interaction is a relevant conservation issue that requires further studies to assess its ecological impact. This work represents the first record of the negative interaction between *O. speciosa* and *M. stellatarum* in Italy.

**Keywords:** introduced plant; alien species; non-indigenous species; *Oenothera speciosa*; stuck proboscis; *Macroglossum stellatarum*; trapped hawkmoths





The genus *Oenothera* (L., 1753), belonging to the family Onagraceae, currently consists of 145 species of flowering plants [1]. These species mainly occur in temperate America as well as in the tropics but also comprise plants native to Europe and established aliens, mostly represented by species escaping from cultivation [2].

Within this genus, *Oenothera speciosa* Nutt. is an herbaceous perennial species native to prairies in the United States of America and northern Mexico [2,3] and is currently a worldwide-distributed species that is widespread in Europe, South America, Asia, and Oceania [4]. This species is one of the most frequently cultivated *Oenothera* species and is listed as escaped from cultivation in many areas of the world, where it is reported as a perennial alien introduced as an ornamental plant [5]. In fact, in the past several years, *O. speciosa* has been expanding its range, and it has recently been reported in many European countries, such as some areas of Spain [6,7], Greece [8], Sweden, Finland [9], Bulgaria [10], and Albania [11], as well as countries in Asia, such as Iraq [12] and China [13].

In Italy, *O. speciosa* has been reported as a casual alien species in Lombardy, Veneto, Tuscany, and Sicily and as a naturalized alien in Marche and Emilia Romagna, and it often grows in abandoned urban gardens, probably introduced a few years ago for ornamental purposes in small flowerbeds, then expanding to the surrounding areas [14,15]. The flowers of this species bloom between May and July and stay open throughout most of the daytime and at night, except for the hottest hours [16].

Many *Oenothera* species are usually pollinated by large nocturnal moths, especially those belonging to the family Sphingidae, and have typical sphingophilous adaptations: a deep hypanthium, pale corolla color, spreading four-lobed stigmas elevated above the anthers, fragrance, and nectar [10]. For this reason, these species are considered "hawkmoth flowers". In fact, sphingid moths have a long proboscis adapted for feeding from specific flowering plants. Although *O. speciosa* shows these features, in its native countries, it is not

usually pollinated by sphingid species but mainly by diurnal insects, such as honeybees, some species of butterflies (e.g., the families Hesperiidae, Papilionidae, and Pieridae), and chrysomelid beetles [16,17]. On the other hand, in countries where *O. species* is listed as an allochthonous species, some sphingid moths were observed feeding on its flowers. In particular, the Eurasian hummingbird hawkmoth *Macroglossum stellatarum* (Linnaeus, 1758), a species native to Europe but also distributed in Asia and northern Africa, is a typical diurnal visitor of *O. speciosa* [10,12].

An interesting negative plant–insect interaction has been documented in France [18,19], Bulgaria [10], and Iraq [12]: some authors have observed that when *M. stellatarum* inserts its proboscis into the hypanthium of the flowers of *O. speciosa*, its tubular mouthpart can become physically stuck in the flower, and the moth remains trapped; the trapped insects generally die after a short period, becoming prey for other faunal organisms.

This work represents the first record of the negative interaction between *O. speciosa* and *M. stellatarum* in Italy (See Supplementary).

The observations were performed in the late spring, from 9 to 11 June 2022, on private land in Brisighella, Ravenna (44.21124 N, 11.78201 E), a small town located in central northern Italy. In this geographic area, *O. speciosa* is considered a naturalized allochthonous species and is sometimes spread aggressively by rhizomes and self-seeding colonies [20]. In the investigated area, several bushes of *O. speciosa* were found near a vineyard, and a total of about 30 flowers were observed. In these few days, a total of nine specimens of *M. stellatarum* were found stuck in the flowers of *O. speciosa*, with the proboscides trapped between the tapered hairs of both the hypanthium and the style (Figure 1). All the specimens were found hovering, exhausted or about to die (Figure 2); many specimens showed damaged wings in an effort to escape, and some of them showed a hairless thorax instead of a scale-covered one (Figure 3). Only two moths self-released in a few minutes when picked up with the flowers. The other specimens were not able to free themselves because their proboscides were too stuck and coiled in the hypanthium. Specimens that were probably trapped for several hours, even though they were helped to break free, showed damaged and unrolled proboscides (Figure 4).

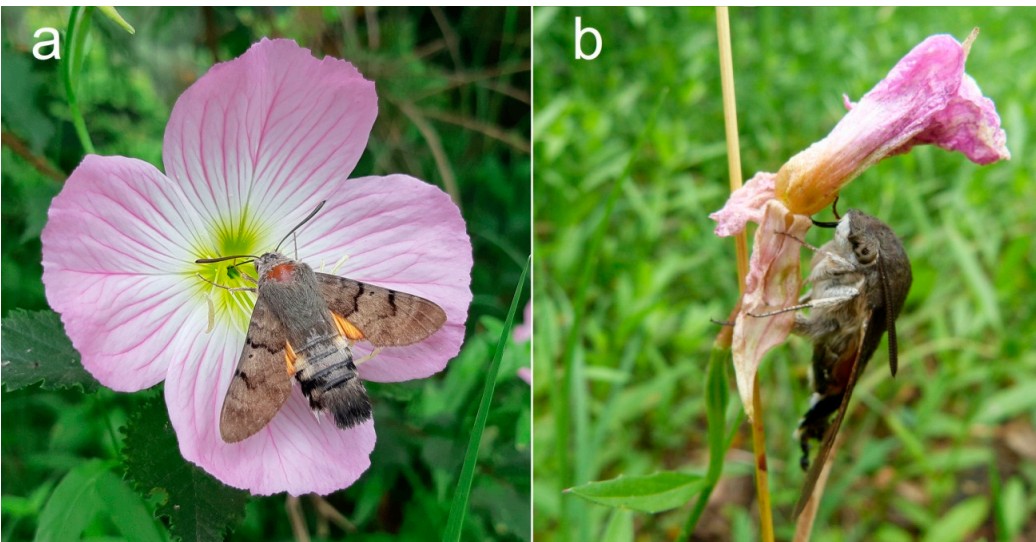

**Figure 1.** Two specimens of the hummingbird hawkmoth *Macroglossum stellatarum* stuck with their proboscides in the flowers of *Oenothera speciosa*. In the first picture (**a**), the flower is newly bloomed, while in the second picture (**b**), the flower is wilted, so it can be hypothesized that the moth was stuck for several hours. Photo credit: Andrea Bonifazi.

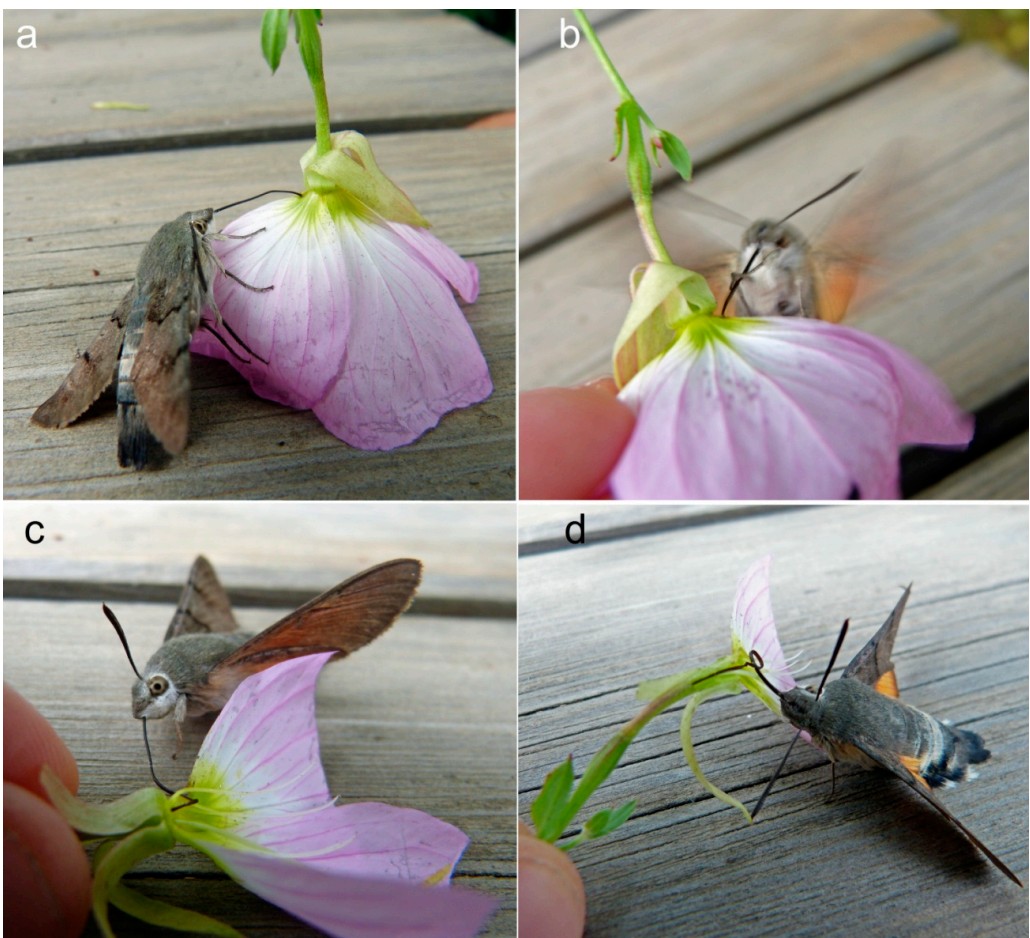

**Figure 2.** *Macroglossum stellatarum* stuck in the flower of *Oenothera speciosa* (**a**) attempting unsuccessfully to free herself by flapping its wings (**b**). However, the attempt made the situation worse, trapping the proboscis even more (**c**). The last picture (**d**) shows the proboscis deeply inserted into the hypanthium of the flower. Photo credit: Andrea Bonifazi.

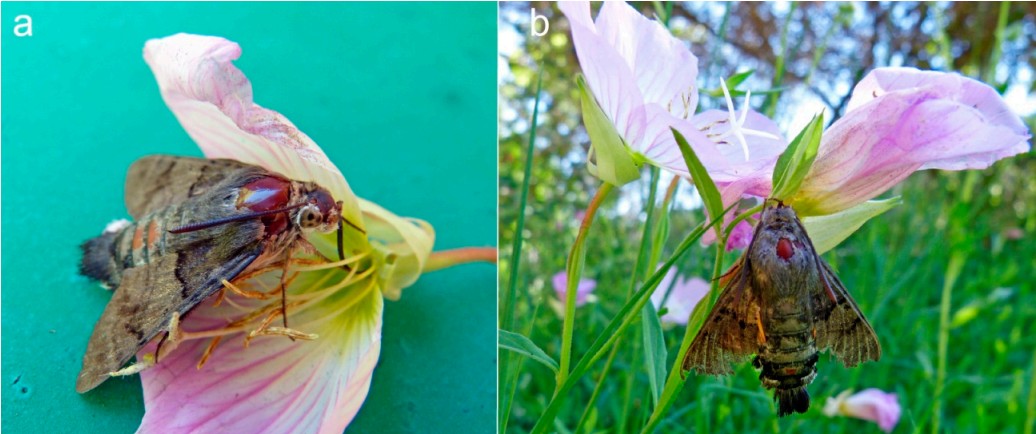

**Figure 3.** Two specimens of *Macroglossum stellatarum* stuck with their proboscides in the flowers of *Oenothera speciosa*; the images show that their wings were very damaged in an effort to escape, and their thoraxes were widely (**a**) or partially (**b**) hairless. Photo credit: Andrea Bonifazi.

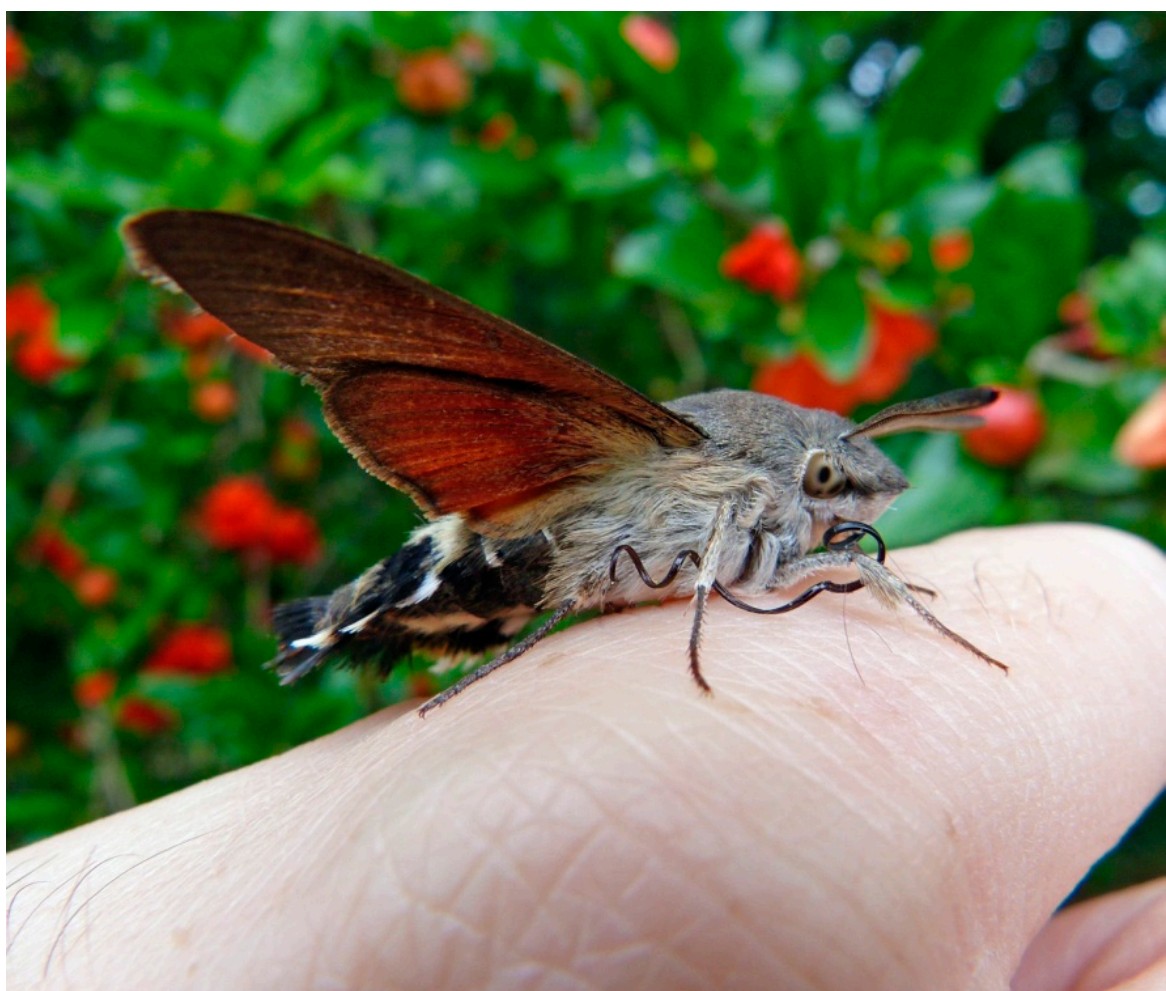

**Figure 4.** An exhausted specimen of *Macroglossum stellatarum* that was stuck in the flower of *Oenothera speciosa* for several hours. The proboscis was unrolled and clearly damaged even though the moths were set free. Photo credit: Andrea Bonifazi.

On the flowers of *O. speciosa*, smaller butterflies belonging to the families Papilionidae and Pieridae were also found, along with diurnal bees, such as *Apis mellifera* Linnaeus, 1758 (Apidae); beetles, such as *Oxythyrea funesta* Poda, 1761 (Scarabaeidae), *Lachnaia italica* Weise, 1881 (Chrysomelidae), *Coccinella septempunctata* Linnaeus, 1758, and *Harmonia axyridis* (Pallas, 1773) (Coccinellidae); and young katydids (Orthoptera Tettigoniidae), but their visits to the flowers appeared to be trouble-free (Figure 5).

As noted by Alkhesraji et al. (2016) in Iraq and Zlatkov et al. (2018) in Bulgaria, other moth species (e.g., *Agrius convolvuli* (Linnaeus, 1758), *Hyles livornica* (Esper, 1780), *Sphinx pinastri* (Linnaeus, 1758), and *Autographa californica* (Speyer, 1875)) were also observed stuck to flowers, but they always escaped without external assistance. It should be noted that some of these moth species are considerably larger than *M. stellatarum* and have longer proboscides. The negative interaction between *O. speciosa* and *M. stellatarum* could be attributed to both the inner morphology of the flower and the structure of the proboscis of this moth: in fact, the style and hypanthium are densely covered with trichomes (Figure 6). The trichomes are non-glandular, unicellular, and oriented toward the base, and they have a thick wall and slightly sigmoid form. As highlighted by Zlatkov et al. (2018), the proboscides of *M. stellatarum* have numerous transverse grooves separating the cuticular annulations, which fit well into the trichome tips; when a moth inserts its proboscis into the flower of *O. speciosa*, the tips of the trichomes enter the grooves, and the reflex movement of the proboscis is hindered by the orientation of the trichomes.

Although in its native range, *O. speciosa* is rarely visited by sphingid moths [16,17], and no cases of dead moths with trapped proboscides have been reported in the literature, in 2017, the species *M. stellatarum* was firstly recorded in California, USA [21]. It might be interesting to know if this negative interaction also occurs in the areas where the plant is indigenous.

To date, we have no data on the impact of *O. speciosa* on *M. stellatarum*, but the killing of an autochthonous insect species by causal or naturalized alien plants requires further studies to assess its ecological impact.

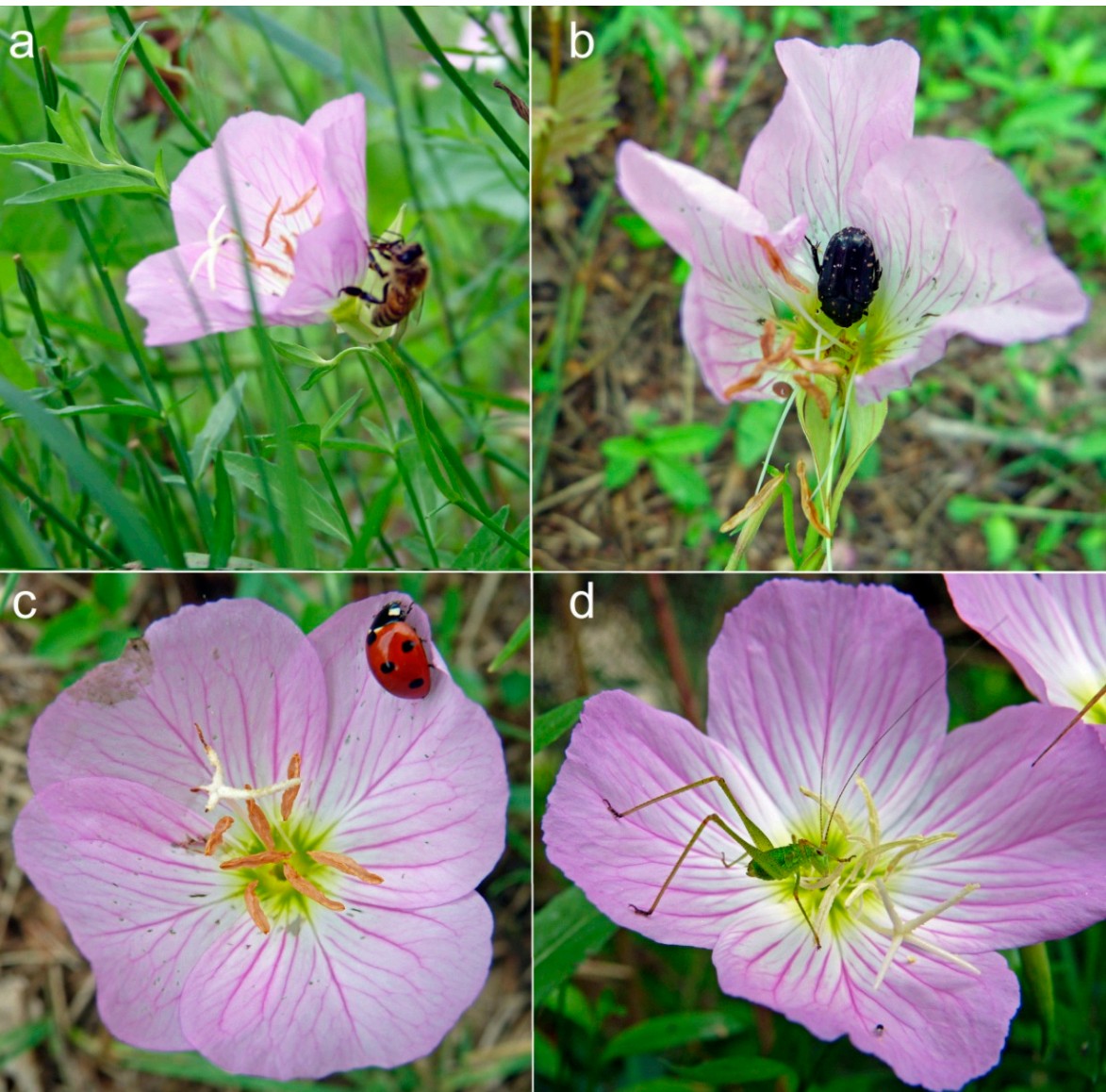

**Figure 5.** Some insect species visiting the flowers *Oenothera speciosa* without showing any damage: (**a**) the honeybee *Apis mellifera* Linnaeus, 1758 (Apidae); the beetles (**b**) *Oxythyrea funesta* Poda, 1761 (Scarabaeidae) and (**c**) *Coccinella septempunctata* Linnaeus, 1758; and (**d**) a young katydid (Orthoptera Tettigoniidae). Photo credit: Andrea Bonifazi.

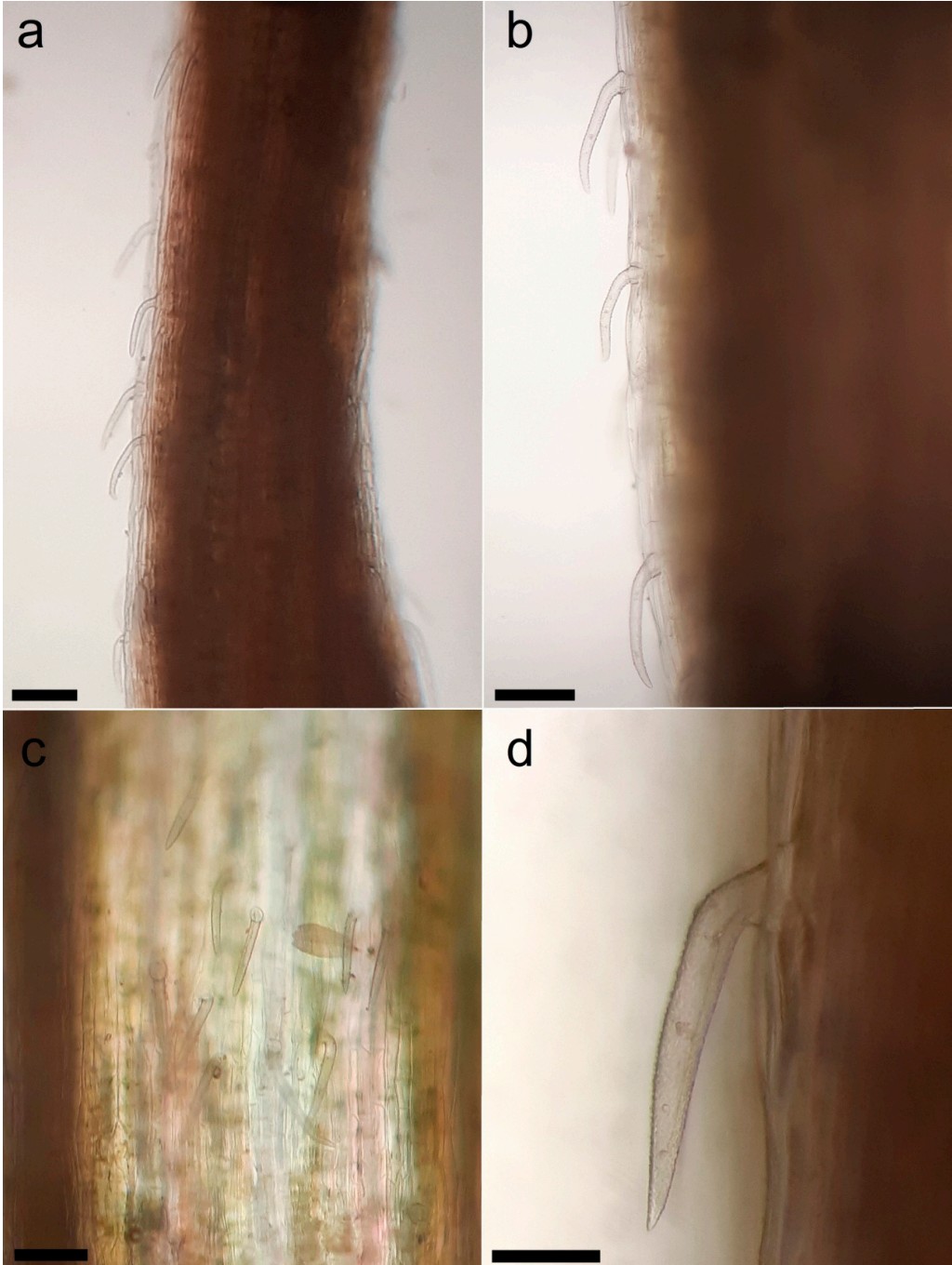

**Figure 6.** Section of the trichomal zone of the style (**a**,**b**) and hypanthium (**c**,**d**) of a flower of *Oenothera speciosa*. It is clear as the internal basal part of the hypanthium and the corresponding part of the style are densely pubescent, and the trichomes are oriented toward the base, with a thick wall and slightly sigmoid form. Scale bars: (**a**,**c**) 100 µm, (**b**) 50 µm, (**d**) 30 µm. Photo credit: Andrea Bonifazi.

**Supplementary Materials:** The following supporting information can be downloaded at: https://www.mdpi.com/article/10.3390/d14090743/s1. *Macroglossum stellatarum* stuck in the flower of *Oenothera speciosa* attempting unsuccessfully to free herself by flapping vigorously its wings. However, the attempt makes the situation worse, trapping the proboscis even more. Video credit: Andrea Bonifazi.

**Author Contributions:** Conceptualization, A.B., M.P. and E.M.; data curation, A.B. and M.P.; methodology, A.B., M.P. and E.M.; investigation, A.B. and M.P.; validation, A.B., M.P. and E.M.; visualization, A.B., M.P. and E.M.; writing—original draft preparation, A.B. writing—review and editing, A.B.; project administration, A.B. All authors have read and agreed to the published version of the manuscript.

**Funding:** This research received no external funding.

**Institutional Review Board Statement:** Not applicable.

**Data Availability Statement:** Not applicable.

**Acknowledgments:** We would like to thank the anonymous reviewers which spent their time to review our manuscript and to enrich it with their comments. We also thank Andrea Gaion for correcting the grammar mistakes and much improving the quality of the text. Finally, we thank Marco Lezzi and Elisabetta Fanti for showing us these places.

**Conflicts of Interest:** The authors declare no conflict of interest.

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
