# Peer review of "The Spreading in Europe of the Non-Indigenous Species *Oenothera speciosa* Nutt. Might Be a Threat to the Autochthonous Moth *Macroglossum stellatarum* (Linnaeus, 1758)? A New Case Study from Italy"

_diversity, doi:10.3390/d14090743_

Round 1

Reviewer 1 Report

The paper presents an interesting interaction between a native moth and an introduced plant, with a negative impact on the moth, in Italy. The manuscript illustrates well the result of this interaction and comments on a probably negative effect on the population of Macroglossum stellatarum.

The text is well written in general, but I noticed numerous grammar mistakes which do not affect the scientific quality of the paper, however. I recommend reading by a native speaker to polish these minor problems.

Author Response

Ref: diversity-1888645
Title: The spreading in Europe of the non indigenous species Oenothera speciosa Nutt. might be a threat to the autochthonous moth Macroglossum stellatarum (Linnaeus, 1758)? A new case study from Italy.
Journal: Diversity (ISSN 1424-2818)

Comments from the editors and reviewers:
We are grateful to have been given the opportunity to revise our manuscript. We carefully considered reviewers comments, and our replies are shown below, explaining how we have revised the paper based on those comments and recommendations. We have reported reviewers comments and in bold our answers.

-Reviewer #1
The paper presents an interesting interaction between a native moth and an introduced plant, with a negative impact on the moth, in Italy. The manuscript illustrates well the result of this interaction and comments on a probably negative effect on the population of Macroglossum stellatarum.
The text is well written in general, but I noticed numerous grammar mistakes which do not affect the scientific quality of the paper, however. I recommend reading by a native speaker to polish these minor problems.

A native English speaker has corrected the grammar mistakes and has much improved the quality of the text.

Reviewer 2 Report

The MS is interesting to read and I upload the authors for this observation. Usually, these kinds of cases are rare or difficult to document in the field. 

The authors indicated that they conducted an observation 9-11, i recommend the authors to indicate the time spent in hours, conducting the observation. 

Perhaps the authors can comment on the presence of antagonists of the moth which could affect the number of the moth trapped.

Author Response

Ref: diversity-1888645
Title: The spreading in Europe of the non indigenous species Oenothera speciosa Nutt. might be a threat to the autochthonous moth Macroglossum stellatarum (Linnaeus, 1758)? A new case study from Italy.
Journal: Diversity (ISSN 1424-2818)

Comments from the editors and reviewers:
We are grateful to have been given the opportunity to revise our manuscript. We carefully considered reviewers comments, and our replies are shown below, explaining how we have revised the paper based on those comments and recommendations. We have reported reviewers comments and in bold our answers.

-Reviewer #2
The MS is interesting to read and I upload the authors for this observation. Usually, these kinds of cases are rare or difficult to document in the field. 
The authors indicated that they conducted an observation 9-11, I recommend the authors to indicate the time spent in hours, conducting the observation. 
Perhaps the authors can comment on the presence of antagonists of the moth which could affect the number of the moth trapped.

According to the reviewer comments, we now specified the time spent by conducting the observations. We added this sentence: "The observations were conducted throughout the day, from mid-morning until early evening. Considering the small area covered by the bushes of O. speciosa, each observation was conducted for about 45 minutes by inspecting each flower". We also specified that no antagonists of M. stellatarum were observed.
Moreover, a native English speaker has corrected the grammar mistakes and has much improved the quality of the text.

Reviewer 3 Report

For the record, this is a common garden plant in California, locally escaping, and I have never seen any insect trapped in the flowers. 

Author Response

Ref: diversity-1888645
Title: The spreading in Europe of the non indigenous species Oenothera speciosa Nutt. might be a threat to the autochthonous moth Macroglossum stellatarum (Linnaeus, 1758)? A new case study from Italy.
Journal: Diversity (ISSN 1424-2818)

Comments from the editors and reviewers:
We are grateful to have been given the opportunity to revise our manuscript. We carefully considered reviewers comments, and our replies are shown below, explaining how we have revised the paper based on those comments and recommendations. We have reported reviewers comments and in bold our answers.

-Reviewer #3
For the record, this is a common garden plant in California, locally escaping, and I have never seen any insect trapped in the flowers.

A native English speaker has corrected the grammar mistakes and has much improved the quality of the text.
Morever, we are grateful for your interesting personal observation. As we specified in the text, it might be interesting to investigate if this negative interaction between O. speciosa and M. stellatarum will occur in the areas where the plant is indigenous.